# Early Convolutions Help Transformers See Better

**Tete Xiao**[1,2]  **Mannat Singh**[1]  **Eric Mintun**[1]  **Trevor Darrell**[2]  **Piotr Dollár**[1*]  **Ross Girshick**[1*]

[1]Facebook AI Research (FAIR)          [2]UC Berkeley

## Abstract

Vision transformer (ViT) models exhibit substandard optimizability. In particular, they are sensitive to the choice of optimizer (AdamW *vs*. SGD), optimizer hyperparameters, and training schedule length. In comparison, modern convolutional neural networks are easier to optimize. Why is this the case? In this work, we conjecture that the issue lies with the *patchify stem* of ViT models, which is implemented by a stride-$p$ $p{\times}p$ convolution ($p = 16$ by default) applied to the input image. This large-kernel plus large-stride convolution runs counter to typical design choices of convolutional layers in neural networks. To test whether this atypical design choice causes an issue, we analyze the optimization behavior of ViT models with their original patchify stem versus a simple counterpart where we replace the ViT stem by a small number of stacked stride-two $3{\times}3$ convolutions. While the vast majority of computation in the two ViT designs is identical, we find that this small change in early visual processing results in markedly different training behavior in terms of the sensitivity to optimization settings as well as the final model accuracy. Using a *convolutional stem* in ViT dramatically increases optimization stability and also improves peak performance (by ∼1-2% top-1 accuracy on ImageNet-1k), while maintaining flops and runtime. The improvement can be observed across the wide spectrum of model complexities (from 1G to 36G flops) and dataset scales (from ImageNet-1k to ImageNet-21k). These findings lead us to recommend using a standard, lightweight convolutional stem for ViT models in this regime as a more robust architectural choice compared to the original ViT model design.

## 1   Introduction

Vision transformer (ViT) models [13] offer an alternative design paradigm to convolutional neural networks (CNNs) [24]. ViTs replace the inductive bias towards local processing inherent in convolutions with global processing performed by multi-headed self-attention [43]. The hope is that this design has the potential to improve performance on vision tasks, akin to the trends observed in natural language processing [11]. While investigating this conjecture, researchers face another unexpected difference between ViTs and CNNs: ViT models exhibit substandard *optimizability*. ViTs are sensitive to the choice of optimizer [41] (AdamW [27] *vs*. SGD), to the selection of dataset specific learning hyperparameters [13, 41], to training schedule length, to network depth [42], *etc*. These issues render former training recipes and intuitions ineffective and impede research.

Convolutional neural networks, in contrast, are exceptionally easy and robust to optimize. Simple training recipes based on SGD, basic data augmentation, and standard hyperparameter values have been widely used for years [19]. Why does this difference exist between ViT and CNN models? In this paper we hypothesize that the issues lies primarily in the *early* visual processing performed by ViT. ViT "patchifies" the input image into $p{\times}p$ non-overlapping patches to form the transformer encoder's input set. This *patchify stem* is implemented as a stride-$p$ $p{\times}p$ convolution, with $p = 16$ as a default value. This large-kernel plus large-stride convolution runs counter to the typical design

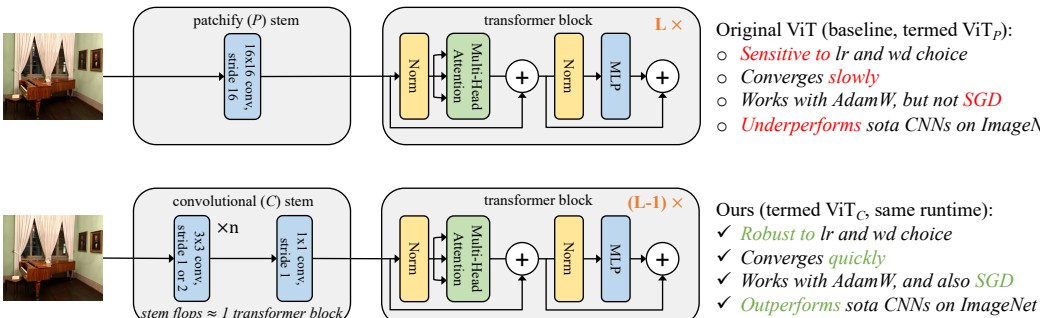

Figure 1: **Early convolutions help transformers see better**: We hypothesize that the substandard optimizability of ViT models compared to CNNs primarily arises from the *early* visual processing performed by its *patchify stem*, which is implemented by a non-overlapping stride-$p$ $p{\times}p$ convolution, with $p = 16$ by default. We *minimally* replace the patchify stem in ViT with a standard *convolutional stem* of only ~5 convolutions that has approximately the same complexity as a *single* transformer block. We reduce the number of transformer blocks by one (*i.e.*, $L - 1$ *vs.* $L$) to maintain parity in flops, parameters, and runtime. We refer to the resulting model as ViT$_C$ and the original ViT as ViT$_P$. The vast majority of computation performed by these two models is identical, yet surprisingly we observe that ViT$_C$ (i) converges faster, (ii) enables, for the first time, the use of either AdamW or SGD without a significant accuracy drop, (iii) shows greater stability to learning rate and weight decay choice, and (iv) yields improvements in ImageNet top-1 error allowing ViT$_C$ to outperform state-of-the-art CNNs, whereas ViT$_P$ does not.

choices used in CNNs, where best-practices have converged to a small stack of stride-two $3{\times}3$ kernels as the network's stem (*e.g.*, [30, 36, 39]).

To test this hypothesis, we *minimally* change the early visual processing of ViT by replacing its patchify stem with a standard *convolutional stem* consisting of only ~5 convolutions, see Figure 1. To compensate for the small addition in flops, we remove one transformer block to maintain parity in flops and runtime. We observe that even though the vast majority of the computation in the two ViT designs is identical, this small change in early visual processing results in markedly different training behavior in terms of the sensitivity to optimization settings as well as the final model accuracy.

In extensive experiments we show that replacing the ViT patchify stem with a more standard convolutional stem (i) allows ViT to converge faster (§5.1), (ii) enables, for the first time, the use of either AdamW or SGD without a significant drop in accuracy (§5.2), (iii) brings ViT's stability w.r.t. learning rate and weight decay closer to that of modern CNNs (§5.3), and (iv) yields improvements in ImageNet [10] top-1 error of ~1-2 percentage points (§6). We consistently observe these improvements across a wide spectrum of model complexities (from 1G flops to 36G flops) and dataset scales (ImageNet-1k to ImageNet-21k).

These results show that injecting some convolutional inductive bias into ViTs can be beneficial under commonly studied settings. We did *not* observe evidence that the hard locality constraint in early layers hampers the representational capacity of the network, as might be feared [9]. In fact we observed the opposite, as ImageNet results improve even with larger-scale models and larger-scale data when using a convolution stem. Moreover, under carefully controlled comparisons, we find that ViTs are only able to surpass state-of-the-art CNNs when equipped with a convolutional stem (§6).

We conjecture that restricting convolutions in ViT to *early* visual processing may be a crucial design choice that strikes a balance between (hard) inductive biases and the representation learning ability of transformer blocks. Evidence comes by comparison to the "hybrid ViT" presented in [13], which uses 40 convolutional layers (most of a ResNet-50) and shows no improvement over the default ViT. This perspective resonates with the findings of [9], who observe that early transformer blocks prefer to learn more local attention patterns than later blocks. Finally we note that exploring the design of hybrid CNN/ViT models is *not* a goal of this work; rather we demonstrate that simply using a minimal convolutional stem with ViT is sufficient to dramatically change its optimization behavior.

In summary, the findings presented in this paper lead us to recommend using a standard, lightweight convolutional stem for ViT models in the analyzed dataset scale and model complexity spectrum as a more robust and higher performing architectural choice compared to the original ViT model design.

## 2   Related Work

**Convolutional neural networks (CNNs).** The breakthrough performance of the AlexNet [23] CNN [15, 24] on ImageNet classification [10] transformed the field of recognition, leading to the development of higher performing architectures, *e.g.*, [19, 36, 37, 48], and scalable training methods [16, 21]. These architectures are now core components in object detection (*e.g.*, [34]), instance segmentation (*e.g.*, [18]), and semantic segmentation (*e.g.*, [26]). CNNs are typically trained with stochastic gradient descent (SGD) and are widely considered to be easy to optimize.

**Self-attention in vision models.** Transformers [43] are revolutionizing natural language processing by enabling scalable training. Transformers use multi-headed self-attention, which performs global information processing and is strictly more general than convolution [6]. Wang *et al.* [46] show that (single-headed) self-attention is a form of non-local means [2] and that integrating it into a ResNet [19] improves several tasks. Ramachandran *et al.* [32] explore this direction further with stand-alone self-attention networks for vision. They report difficulties in designing an attention-based network stem and present a bespoke solution that avoids convolutions. In contrast, we demonstrate the benefits of a convolutional stem. Zhao *et al.* [53] explore a broader set of self-attention operations with hard-coded locality constraints, more similar to standard CNNs.

**Vision transformer (ViT).** Dosovitskiy *et al.* [13] apply a transformer encoder to image classification with minimal vision-specific modifications. As the counterpart of input token embeddings, they partition the input image into, *e.g.*, 16×16 pixel, non-overlapping patches and linearly project them to the encoder's input dimension. They report lackluster results when training on ImageNet-1k, but demonstrate state-of-the-art transfer learning when using large-scale pretraining data. ViTs are sensitive to many details of the training recipe, *e.g.*, they benefit greatly from AdamW [27] compared to SGD and require careful learning rate and weight decay selection. ViTs are generally considered to be difficult to optimize compared to CNNs (*e.g.*, see [13, 41, 42]). Further evidence of challenges comes from Chen *et al.* [4] who report ViT optimization instability in self-supervised learning (unlike with CNNs), and find that freezing the patchify stem at its random initialization improves stability.

**ViT improvements.** ViTs are gaining rapid interest in part because they may offer a novel direction away from CNNs. Touvron *et al.* [41] show that with more regularization and stronger data augmentation ViT models achieve competitive accuracy on ImageNet-1k alone (*cf*. [13]). Subsequently, works concurrent with our own explore numerous other ViT improvements. Dominant themes include multi-scale networks [14, 17, 25, 45, 50], increasing depth [42], and locality priors [5, 9, 17, 47, 49]. In [9], d'Ascoli *et al.* modify multi-head self-attention with a convolutional bias at initialization and show that this prior improves sample efficiency and ImageNet accuracy. Resonating with our work, [5, 17, 47, 49] present models with convolutional stems, but do not analyze optimizability (our focus).

**Discussion.** Unlike the concurrent work on locality priors in ViT, our focus is studying *optimizability* under *minimal* ViT modifications in order to derive crisp conclusions. Our perspective brings several novel observations: by adding only ~5 convolutions to the stem, ViT can be optimized well with either AdamW or SGD (*cf*. all prior works use AdamW to avoid large drops in accuracy [41]), it becomes less sensitive to the specific choice of learning rate and weight decay, and training converges faster. We also observe a consistent improvement in ImageNet top-1 accuracy across a wide spectrum of model complexities (1G flops to 36G flops) and dataset scales (ImageNet-1k to ImageNet-21k). These results suggest that a (hard) convolutional bias early in the network does not compromise representational capacity, as conjectured in [9], and is beneficial within the scope of this study.

## 3   Vision Transformer Architectures

Next, we review vision transformers [13] and describe the convolutional stems used in our work.

**The vision transformer (ViT).** ViT first partitions an input image into *non-overlapping* $p \times p$ patches and linearly projects each patch to a $d$-dimensional feature vector using a learned weight matrix. A patch size of $p = 16$ and an image size of 224×224 are typical. The resulting patch embeddings (plus positional embeddings and a learned classification token embedding) are processed by a standard transformer encoder [43, 44] followed by a classification head. Using common network nomenclature, we refer to the portion of ViT before the transformer blocks as the network's *stem*. ViT's stem is a

| model | ref model | hidden size | MLP mult | num heads | num blocks | flops (B) | params (M) | acts (M) | time (min) | model | hidden size | MLP mult | num heads | num blocks | flops (B) | params (M) | acts (M) | time (min) |
|---|---|---|---|---|---|---|---|---|---|---|---|---|---|---|---|---|---|---|
| $ViT_P$-1GF | ∼ViT-T | 192 | 3 | 3 | 12 | 1.1 | 4.8 | 5.5 | 2.6 | $ViT_C$-1GF | 192 | 3 | 3 | 11 | 1.1 | 4.6 | 5.7 | 2.7 |
| $ViT_P$-4GF | ∼ViT-S | 384 | 3 | 6 | 12 | 3.9 | 18.5 | 11.1 | 3.8 | $ViT_C$-4GF | 384 | 3 | 6 | 11 | 4.0 | 17.8 | 11.3 | 3.9 |
| $ViT_P$-18GF | =ViT-B | 768 | 4 | 12 | 12 | 17.5 | 86.7 | 24.0 | 11.5 | $ViT_C$-18GF | 768 | 4 | 12 | 11 | 17.7 | 81.6 | 24.1 | 11.4 |
| $ViT_P$-36GF | $\frac{3}{5}$ViT-L | 1024 | 4 | 16 | 14 | 35.9 | 178.4 | 37.3 | 18.8 | $ViT_C$-36GF | 1024 | 4 | 16 | 13 | 35.0 | 167.8 | 36.7 | 18.6 |

Table 1: **Model definitions**: *Left*: Our $ViT_P$ models at various complexities, which use the original *patchify* stem and closely resemble the original ViT models [13]. To facilitate comparisons with CNNs, we modify the original ViT-Tiny, -Small, -Base, -Large models to obtain models at 1GF, 4GF, 18GF, and 36GF, respectively. The modifications are indicated in blue and include reducing the MLP multiplier from 4× to 3× for the 1GF and 4GF models, and reducing the number of transformer blocks from 24 to 14 for the 36GF model. *Right*: Our $ViT_C$ models at various complexities that use the *convolutional* stem. The only additional modification relative to the corresponding $ViT_P$ models is the removal of 1 transformer block to compensate for the increased flops of the convolutional stem. We show complexity measures for all models (flops, parameters, activations, and epoch training time on ImageNet-1k); the corresponding $ViT_P$ and $ViT_C$ models match closely on all metrics.

specific case of convolution (stride-$p$, $p{\times}p$ kernel), but we will refer to it as the *patchify stem* and reserve the terminology of *convolutional stem* for stems with a more conventional CNN design with multiple layers of *overlapping* convolutions (*i.e.*, with stride smaller than the kernel size).

**$ViT_P$ models.** Prior work proposes ViT models of various sizes, such as ViT-Tiny, ViT-Small, ViT-Base, *etc*. [13, 41]. To facilitate comparisons with CNNs, which are typically standardized to 1 gigaflop (GF), 2GF, 4GF, 8GF, *etc*., we modify the original ViT models to obtain models at about these complexities. Details are given in Table 1 (left). For easier comparison with CNNs of similar flops, and to avoid subjective size names, we refer the models by their flops, *e.g.*, $ViT_P$-4GF in place of ViT-Small. We use the $P$ subscript to indicate that these models use the original *patchify* stem.

**Convolutional stem design.** We adopt a typical minimalist convolutional stem design by stacking 3×3 convolutions [36], followed by a single 1×1 convolution at the end to match the $d$-dimensional input of the transformer encoder. These stems quickly downsample a 224×224 input image using overlapping strided convolutions to 14×14, matching the number of inputs created by the standard patchify stem. We follow a simple design pattern: all 3×3 convolutions either have stride 2 and double the number of output channels or stride 1 and keep the number of output channels constant. We enforce that the stem accounts for approximately the computation of one transformer block of the corresponding model so that we can easily control for flops by removing one transformer block when using the convolutional stem instead of the patchify stem. Our stem design was chosen to be purposefully simple and we emphasize that it was not designed to maximize model accuracy.

**$ViT_C$ models.** To form a ViT model with a convolutional stem, we simply replace the patchify stem with its counterpart convolutional stem and *remove one transformer block* to compensate for the convolutional stem's extra flops (see Figure 1). We refer to the modified ViT with a convolutional stem as $ViT_C$. Configurations for $ViT_C$ at various complexities are given in Table 1 (right); corresponding $ViT_P$ and $ViT_C$ models match closely on all complexity metrics including flops and runtime.

**Convolutional stem details.** Our convolutional stem designs use four, four, and six 3×3 convolutions for the 1GF, 4GF, and 18GF models, respectively. The output channels are [24, 48, 96, 192], [48, 96, 192, 384], and [64, 128, 128, 256, 256, 512], respectively. All 3×3 convolutions are followed by batch norm (BN) [21] and then ReLU [29], while the final 1×1 convolution is not, to be consistent with the original patchify stem. Eventually, matching stem flops to transformer block flops results in an unreasonably large stem, thus $ViT_C$-36GF uses the same stem as $ViT_C$-18GF.

**Convolutions in ViT.** Dosovitskiy *et al*. [13] also introduced a "hybrid ViT" architecture that blends a modified ResNet [19] (BiT-ResNet [22]) with a transformer encoder. In their hybrid model, the patchify stem is replaced by a partial BiT-ResNet-50 that terminates at the output of the conv4 stage or the output of an extended conv3 stage. These image embeddings replace the standard patchify stem embeddings. This partial BiT-ResNet-50 stem is *deep*, with 40 convolutional layers. In this work, we explore *lightweight* convolutional stems that consist of only 5 to 7 convolutions in total, instead of the 40 used by the hybrid ViT. Moreover, we emphasize that the goal of our work is *not* to explore the hybrid ViT design space, but rather to study the optimizability effects of simply replacing the patchify stem with a *minimal* convolutional stem that follows standard CNN design practices.

# 4    Measuring Optimizability

It has been noted in the literature that ViT models are challenging to optimize, *e.g.*, they may achieve only modest performance when trained on a mid-size dataset (ImageNet-1k) [13], are sensitive to data augmentation [41] and optimizer choice [41], and may perform poorly when made deeper [42]. We empirically observed the general presence of such difficulties through the course of our experiments and informally refer to such optimization characteristics collectively as *optimizability*.

Models with poor optimizability can yield very different results when hyperparameters are varied, which can lead to seemingly bizarre observations, *e.g.*, removing *erasing* data augmentation [54] causes a catastrophic drop in ImageNet accuracy in [41]. Quantitative metrics to measure optimizability are needed to allow for more robust comparisons. In this section, we establish the foundations of such comparisons; we extensively test various models using these optimizability measures in §5.

**Training length stability.** Prior works train ViT models for lengthy schedules, *e.g.*, 300 to 400 epochs on ImageNet is typical (at the extreme, [17] trains models for 1000 epochs), since results at a formerly common 100-epoch schedule are substantially worse (2-4% lower top-1 accuracy, see §5.1). In the context of ImageNet, we define top-1 accuracy at 400 epochs as an approximate asymptotic result, *i.e.*, training for longer will not meaningfully improve top-1 accuracy, and we compare it to the accuracy of models trained for only 50, 100, or 200 epochs. We define *training length stability* as the gap to asymptotic accuracy. Intuitively, it's a measure of convergence speed. Models that converge faster offer obvious practical benefits, especially when training many model variants.

**Optimizer stability.** Prior works use AdamW [27] to optimize ViT models from random initialization. Results of SGD are not typically presented and we are only aware of Touvron *et al.* [41]'s report of a dramatic ~7% drop in ImageNet top-1 accuracy. In contrast, widely used CNNs, such as ResNets, can be optimized equally well with either SGD or AdamW (see §5.2) and SGD (always with momentum) is typically used in practice. SGD has the practical benefit of having fewer hyperparameters (*e.g.*, tuning AdamW's $\beta_2$ can be important [3]) and requiring 50% less optimizer state memory, which can ease scaling. We define *optimizer stability* as the accuracy gap between AdamW and SGD. Like training length stability, we use optimizer stability as a proxy for the ease of optimization of a model.

**Hyperparameter (*lr*, *wd*) stability.** Learning rate (*lr*) and weight decay (*wd*) are among the most important hyperparameters governing optimization with SGD and AdamW. New models and datasets often require a search for their optimal values as the choice can dramatically affect results. It is desirable to have a model and optimizer that yield good results for a wide range of learning rate and weight decay values. We will explore this *hyperparameter stability* by comparing the error distribution functions (EDFs) [30] of models trained with various choices of *lr* and *wd*. In this setting, to create an EDF for a model we randomly sample values of *lr* and *wd* and train the model accordingly. Distributional estimates, like those provided by EDFs, give a more complete view of the characteristics of models that point estimates cannot reveal [30, 31]. We will review EDFs in §5.3.

**Peak performance.** The maximum possible performance of each model is the most commonly used metric in previous literature and it is often provided without carefully controlling training details such as data augmentations, regularization methods, number of epochs, and *lr*, *wd* tuning. To make more robust comparisons, we define *peak performance* as the result of a model at 400 epochs using its best-performing optimizer and *parsimoniously* tuned *lr* and *wd* values (details in §6), *while fixing justifiably good values for all other variables that have a known impact on training*. Peak performance results for ViTs and CNNs under these carefully controlled training settings are presented in §6.

# 5    Stability Experiments

In this section we test the *stability* of ViT models with the original patchify ($P$) stem *vs.* the convolutional ($C$) stem defined in §3. For reference, we also train RegNetY [12, 31], a state-of-the-art CNN that is easy to optimize and serves as a reference point for good stability.

We conduct experiments using ImageNet-1k [10]'s standard training and validation sets, and report top-1 error. Following [12], for all results, we carefully control training settings and we use a minimal set of data augmentations that still yields strong results, for details see §5.4. In this section, unless noted, for each model we use the optimal *lr* and *wd* found under a 50 epoch schedule (see Appendix).

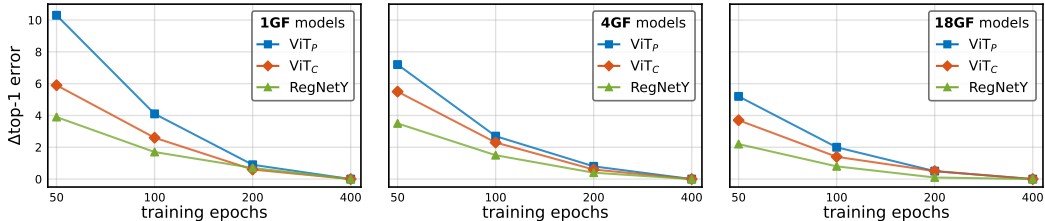

Figure 2: **Training length stability**: We train 9 models for 50 to 400 epochs on ImageNet-1k and plot the $\Delta$top-1 error to the 400 epoch result for each. ViT$_C$ demonstrates faster convergence than ViT$_P$ across the model complexity spectrum, and helps close the gap to CNNs (represented by RegNetY).

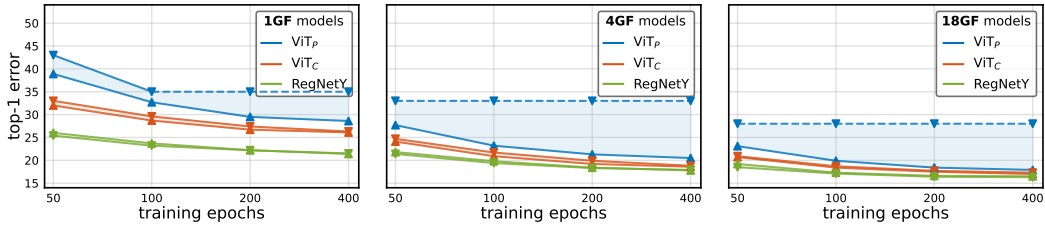

Figure 3: **Optimizer stability**: We train each model for 50 to 400 epochs with AdamW (upward triangle ▲) and SGD (downward triangle ▼). For the baseline ViT$_P$, SGD yields significantly worse results than AdamW. In contrast, ViT$_C$ and RegNetY models exhibit a much smaller gap between SGD and AdamW across all settings. Note that for long schedules, ViT$_P$ often fails to converge with SGD (*i.e.*, loss goes to NaN), in such cases we copy the best results from a shorter schedule of the same model (and show the results via a dashed line).

## 5.1 Training Length Stability

We first explore how rapidly networks converge to their asymptotic error on ImageNet-1k, *i.e.*, the highest possible accuracy achievable by training for many epochs. We approximate asymptotic error as a model's error using a 400 epoch schedule based on observing diminishing returns from 200 to 400. We consider a grid of 24 experiments for ViT: $\{P, C\}$ stems $\times$ $\{1, 4, 18\}$ GF model sizes $\times$ $\{50, 100, 200, 400\}$ epochs. For reference we also train RegNetY at $\{1, 4, 16\}$ GF. We use the best optimizer choice for each model (AdamW for ViT models and SGD for RegNetY models).

**Results.** Figure 2 shows the absolute error *deltas* ($\Delta$top-1) between 50, 100, and 200 epoch schedules and asymptotic performance (at 400 epochs). ViT$_C$ demonstrates faster convergence than ViT$_P$ across the model complexity spectrum, and closes much of the gap to the rate of CNN convergence. The improvement is most significant in the shortest training schedule (50 epoch), *e.g.*, ViT$_P$-1GF has a 10% error delta, while ViT$_C$-1GF reduces this to about 6%. This opens the door to applications that execute a large number of short-scheduled experiments, such as neural architecture search.

## 5.2 Optimizer Stability

We next explore how well AdamW and SGD optimize ViT models with the two stem types. We consider the following grid of 48 ViT experiments: $\{P, C\}$ stems $\times$ $\{1, 4, 18\}$ GF sizes $\times$ $\{50, 100, 200, 400\}$ epochs $\times$ $\{$AdamW, SGD$\}$ optimizers. As a reference, we also train 24 RegNetY baselines, one for each complexity regime, epoch length, and optimizer.

**Results.** Figure 3 shows the results. As a baseline, RegNetY models show virtually no gap when trained using either SGD or AdamW (the difference $\sim$0.1-0.2% is within noise). On the other hand, *ViT$_P$ models suffer a dramatic drop when trained with SGD* across all settings (of up to 10% for larger models and longer training schedules). With a convolutional stem, ViT$_C$ models exhibit much smaller error gaps between SGD and AdamW across all training schedules and model complexities, including in larger models and longer schedules, where the gap is reduced to less than 0.2%. In other words, both RegNetY and ViT$_C$ can be easily trained via either SGD or AdamW, but ViT$_P$ cannot.

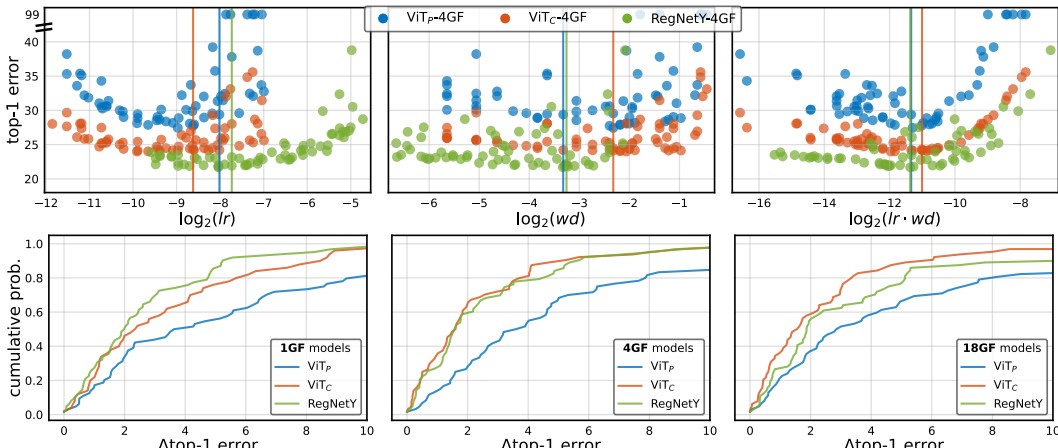

Figure 4: **Hyperparameter stability for AdamW (*lr* and *wd*)**: For each model, we train 64 instances of the model for 50 epochs each with a random *lr* and *wd* (in a fixed width interval around the optimal value for each model). *Top*: Scatterplots of the *lr*, *wd*, and *lr ·wd* for three 4GF models. Vertical bars indicate optimal *lr*, *wd*, and *lr ·wd* values for each model. *Bottom*: For each model, we generate an EDF of the errors by plotting the cumulative distribution of the Δtop-1 errors (Δ to the optimal error for each model). A steeper EDF indicates better stability to *lr* and *wd* variation. ViT$_C$ significantly improves the stability over the baseline ViT$_P$ across the model complexity spectrum, and matches or even outperforms the stability of the CNN model (RegNetY).

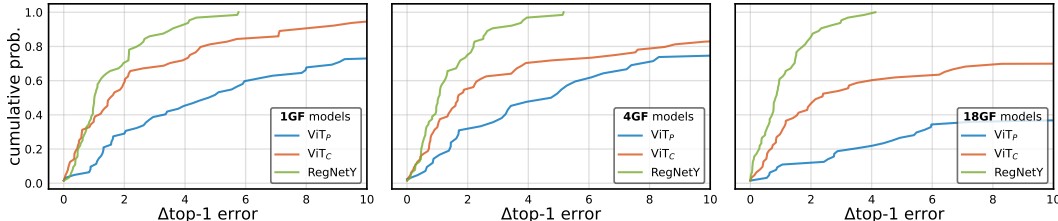

Figure 5: **Hyperparameter stability for SGD (*lr* and *wd*)**: We repeat the setup from Figure 4 using SGD instead of AdamW. The stability improvement of ViT$_C$ over the baseline ViT$_P$ is even larger than with AdamW. *E.g.*, ~60% of ViT$_C$-18GF models are within 4% Δtop-1 error of the best result, while less than 20% of ViT$_P$-18GF models are (in fact most ViT$_P$-18GF runs don't converge).

### 5.3 Learning Rate and Weight Decay Stability

Next, we characterize how sensitive different model families are to changes in learning rate (*lr*) and weight decay (*wd*) under both AdamW and SGD optimizers. To quantify this, we make use of error distribution functions (EDFs) [30]. An EDF is computed by sorting a set of results from low-to-high error and plotting the cumulative proportion of results as error increases, see [30] for details. In particular, we generate EDFs of a model as a function of *lr* and *wd*. The intuition is that if a model is robust to these hyperparameter choices, the EDF will be steep (all models will perform similarly), while if the model is sensitive, the EDF will be shallow (performance will be spread out).

We test 6 ViT models ($\{P, C\} \times \{1, 4, 18\}$ GF) and 3 RegNetY models ($\{1, 4, 16\}$ GF). For each model and each optimizer, we compute an EDF by randomly sampling 64 (*lr*, *wd*) pairs with learning rate and weight decay sampled in a fixed width interval around their optimal values for that model and optimizer (see the Appendix for sampling details). Rather than plotting absolute error in the EDF, we plot Δtop-1 error between the best result (obtained with the optimal *lr* and *wd*) and the observed result. Due to the large number of models, we train each for only 50 epochs.

**Results.** Figure 4 shows scatterplots and EDFs for models trained by AdamW. Figure 5 shows SGD results. In all cases we see that ViT$_C$ significantly improves the *lr* and *wd* stability over ViT$_P$ for both optimizers. This indicates that the *lr* and *wd* are easier to optimize for ViT$_C$ than for ViT$_P$.

## 5.4 Experimental Details

In all experiments we train with a single half-period cosine learning rate decay schedule with a 5-epoch linear learning rate warm-up [16]. We use a minibatch size of 2048. Crucially, weight decay is *not* applied to the gain factors found in normalization layers nor to bias parameters anywhere in the model; we found that decaying these parameters can dramatically reduce top-1 accuracy for small models and short schedules. For inference, we use an exponential moving average (EMA) of the model weights (*e.g.*, [8]). The *lr* and *wd* used in this section are reported in the Appendix. Other hyperparameters use defaults: SGD momentum is 0.9 and AdamW's $\beta_1 = 0.9$ and $\beta_2 = 0.999$.

**Regularization and data augmentation.** We use a simplified training recipe compared to recent work such as DeiT [41], which we found to be equally effective across a wide spectrum of model complexities and dataset scales. We use AutoAugment [7], mixup [52] ($\alpha = 0.8$), CutMix [51] ($\alpha = 1.0$), and label smoothing [38] ($\epsilon = 0.1$). We prefer this setup because it is similar to common settings for CNNs (*e.g.*, [12]) except for stronger mixup and the addition of CutMix (ViTs benefit from both, while CNNs are not harmed). We compare this recipe to the one used for DeiT models in the Appendix, and observe that *our setup provides substantially faster training convergence* likely because we remove repeating augmentation [1, 20], which is known to slow training [1].

# 6 Peak Performance

A model's peak performance is the most commonly used metric in network design. It represents what is possible with the best-known-so-far settings and naturally evolves over time. Making fair comparisons between different models is desirable but fraught with difficulty. Simply citing results from prior work may be negatively biased against that work as it was unable to incorporate newer, yet applicable improvements. Here, we strive to provide a *fairer comparison* between state-of-the-art CNNs, $ViT_P$, and $ViT_C$. We identify a set of factors and then strike a pragmatic balance between which subset to optimize for each model *vs.* which subset share a constant value across all models.

In our comparison, all models share the same epochs (400), use of model weight EMA, and set of regularization and augmentation methods (as specified in §5.4). All CNNs are trained with SGD with *lr* of 2.54 and *wd* of 2.4e−5; we found this single choice worked well across all models, as similarly observed in [12]. For all ViT models we found AdamW with a *lr/wd* of 1.0e−3/0.24 was effective, except for the 36GF models. For these larger models we tested a few settings and found a *lr/wd* of 6.0e−4/0.28 to be more effective for both $ViT_P$-36GF and $ViT_C$-36GF models. For training and inference, ViTs use 224×224 resolution (we do *not* fine-tune at higher resolutions), while the CNNs use (often larger) optimized resolutions specified in [12, 39]. Given this protocol, we compare $ViT_P$, $ViT_C$, and CNNs across a spectrum of model complexities (1GF to 36GF) and dataset scales (directly training on ImageNet-1k *vs.* pretraining on ImageNet-21k and then fine-tuning on ImageNet-1k).

**Results.** Figure 6 shows a progression of results. Each plot shows ImageNet-1k val top-1 error *vs.* ImageNet-1k epoch training time.[1] The left plot compares several state-of-the-art CNNs. RegNetY and RegNetZ [12] achieve similar results across the training speed spectrum and outperform EfficientNets [39]. Surprisingly, ResNets [19] are highly competitive at fast runtimes, showing that under a fairer comparison these years-old models perform substantially better than often reported (*cf*. [39]).

The middle plot compares two representative CNNs (ResNet and RegNetY) to ViTs, still using only ImageNet-1k training. The baseline $ViT_P$ underperforms RegNetY across the entire model complexity spectrum. To our surprise, *$ViT_P$ also underperforms ResNets* in this regime. $ViT_C$ is more competitive and outperforms CNNs in the middle-complexity range.

The right plot compares the same models but with ImageNet-21k pretraining (details in Appendix). In this setting ViT models demonstrates a greater capacity to benefit from the larger-scale data: now $ViT_C$ strictly outperforms both $ViT_P$ and RegNetY. Interestingly, *the original $ViT_P$ does not outperform a state-of-the-art CNN* even when trained on this much larger dataset. Numerical results are presented in Table 2 for reference to exact values. This table also highlights that flop counts are not significantly correlated with runtime, but that activations are (see Appendix for more details), as also observed by [12]. *E.g.*, EfficientNets are slow relative to their flops while ViTs are fast.

---

[1]We time models in PyTorch on 8 32GB Volta GPUs. We note that batch inference time is highly correlated with training time, but we report epoch time as it is easy to interpret and does not depend on the use case.

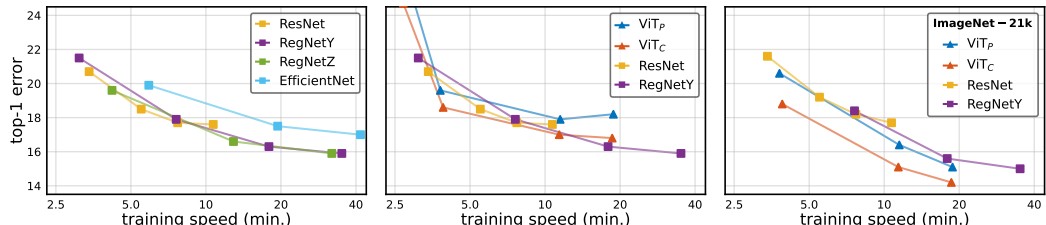

Figure 6: **Peak performance (epoch training time *vs*. ImageNet-1k val top-1 error)**: Results of a fair, controlled comparison of $ViT_P$, $ViT_C$, and CNNs. Each curve corresponds to a model complexity sweep resulting in a training speed spectrum (minutes per ImageNet-1k epoch). *Left:* State-of-the-art CNNs. Equipped with a modern training recipe, ResNets are highly competitive in the faster regime, while RegNetY and Z perform similarly, and better than EfficientNets. *Middle:* Selected CNNs compared to ViTs. With access to only ImageNet-1k training data, RegNetY *and* ResNet outperform $ViT_P$ across the board. $ViT_C$ is more competitive with CNNs. *Right:* Pretraining on ImageNet-21k improves the ViT models more than the CNNs, making $ViT_P$ competitive. Here, the proposed $ViT_C$ outperforms all other models across the full training speed spectrum.

| model | flops (B) | params (M) | acts (M) | time (min) | batch size | 100 | 200 | 400 | IN 21k |
|---|---|---|---|---|---|---|---|---|---|
| ResNet-50 | 4.1 | 25.6 | 11.3 | 3.4 | 2048 | 22.5 | 21.2 | 20.7 | 21.6 |
| ResNet-101 | 7.8 | 44.5 | 16.4 | 5.5 | 2048 | 20.3 | 19.1 | 18.5 | 19.2 |
| ResNet-152 | 11.5 | 60.2 | 22.8 | 7.7 | 2048 | 19.5 | 18.4 | 17.7 | 18.2 |
| ResNet-200 | 15.0 | 64.7 | 32.3 | 10.7 | 1024 | 19.5 | 18.3 | 17.6 | 17.7 |
| RegNetY-1GF | 1.0 | 9.6 | 6.2 | 3.1 | 2048 | 23.2 | 22.2 | 21.5 | - |
| RegNetY-4GF | 4.1 | 22.4 | 14.5 | 7.6 | 2048 | 19.4 | 18.3 | 17.9 | 18.4 |
| RegNetY-16GF | 15.5 | 72.3 | 30.7 | 17.9 | 1024 | 17.1 | 16.4 | 16.3 | 15.6 |
| RegNetY-32GF | 31.1 | 128.6 | 46.2 | 35.1 | 512 | 16.2 | 15.9 | 15.9 | 15.0 |
| RegNetZ-1GF | 1.0 | 11.0 | 8.8 | 4.2 | 2048 | 20.8 | 20.2 | 19.6 | - |
| RegNetZ-4GF | 4.0 | 28.1 | 24.3 | 12.9 | 1024 | 17.4 | 16.9 | 16.6 | - |
| RegNetZ-16GF | 16.0 | 95.3 | 51.3 | 32.0 | 512 | 16.0 | 15.9 | 15.9 | - |
| RegNetZ-32GF | 32.0 | 175.1 | 79.6 | 55.3 | 256 | 16.3 | 16.2 | 16.1 | - |
| EffNet-B2 | 1.0 | 9.1 | 13.8 | 5.9 | 2048 | 21.4 | 20.5 | 19.9 | - |
| EffNet-B4 | 4.4 | 19.3 | 49.5 | 19.4 | 512 | 18.5 | 17.8 | 17.5 | - |
| EffNet-B5 | 10.3 | 30.4 | 98.9 | 41.7 | 256 | 17.3 | 17.0 | 17.0 | - |
| $ViT_P$-1GF | 1.1 | 4.8 | 5.5 | 2.6 | 2048 | 33.2 | 29.7 | 27.7 | - |
| $ViT_P$-4GF | 3.9 | 18.5 | 11.1 | 3.8 | 2048 | 23.3 | 20.8 | 19.6 | 20.6 |
| $ViT_P$-18GF | 17.5 | 86.6 | 24.0 | 11.5 | 1024 | 19.9 | 18.4 | 17.9 | 16.4 |
| $ViT_P$-36GF | 35.9 | 178.4 | 37.3 | 18.8 | 512 | 19.9 | 18.8 | 18.2 | 15.1 |
| $ViT_C$-1GF | 1.1 | 4.6 | 5.7 | 2.7 | 2048 | 28.6 | 26.1 | 24.7 | - |
| $ViT_C$-4GF | 4.0 | 17.8 | 11.3 | 3.9 | 2048 | 20.9 | 19.2 | 18.6 | 18.8 |
| $ViT_C$-18GF | 17.7 | 81.6 | 24.1 | 11.4 | 1024 | 18.4 | 17.5 | 17.0 | 15.1 |
| $ViT_C$-36GF | 35.0 | 167.8 | 36.7 | 18.6 | 512 | 18.3 | 17.6 | 16.8 | 14.2 |

Table 2: **Peak performance (grouped by model family)**: Model complexity and validation top-1 error at 100, 200, and 400 epoch schedules on ImageNet-1k, and the top-1 error after pretraining on ImageNet-21k (IN 21k) and fine-tuning on ImageNet-1k. This table serves as reference for the results shown in Figure 6. Blue numbers: best model trainable under 20 minutes per ImageNet-1k epoch. Batch sizes and training times are reported normalized to 8 32GB Volta GPUs (see Appendix). Additional results on the ImageNet-V2 [33] test set are presented in the Appendix.

These results verify that $ViT_C$'s convolutional stem improves not only optimization stability, as seen in the previous section, but also peak performance. Moreover, this benefit can be seen across the model complexity and dataset scale spectrum. Perhaps surprisingly, given the recent excitement over ViT, we find that $ViT_P$ struggles to compete with state-of-the-art CNNs. We only observe improvements over CNNs when using *both* large-scale pretraining data *and* the proposed convolutional stem.

## 7 Conclusion

In this work we demonstrated that the optimization challenges of ViT models are linked to the large-stride, large-kernel convolution in ViT's patchify stem. The seemingly trivial change of replacing this patchify stem with a simple convolutional stem leads to a remarkable change in optimization behavior. With the convolutional stem, ViT (termed $ViT_C$) converges faster than the original ViT (termed $ViT_P$) (§5.1), trains well with either AdamW or SGD (§5.2), improves learning rate and weight decay stability (§5.3), and improves ImageNet top-1 error by ~1-2% (§6). These results are consistent across a wide spectrum of model complexities (1GF to 36GF) and dataset scales (ImageNet-1k to ImageNet-21k). Our results indicate that injecting a small dose of convolutional inductive bias into the early stages of ViTs can be hugely beneficial. Looking forward, we are interested in the theoretical foundation of why such a minimal architectural modification can have such large (positive) impact on optimizability. We are also interested in studying larger models. Our preliminary explorations into 72GF models reveal that the convolutional stem still improves top-1 error, however we also find that a *new* form of instability arises that causes training error to randomly spike, especially for $ViT_C$.

**Acknowledgements.** We thank Hervé Jegou, Hugo Touvron, and Kaiming He for valuable feedback.

