| stem | kernel size | stride | padding | channels | flops (M) | params (M) | acts (M) | top-1 error AdamW | SGD | Δ |
|------|-------------|--------|---------|----------|-----------|------------|----------|-------------------|-----|---|
| $P$ | [16] | [16] | [0] | [384] | 58 | 0.3 | 0.8 | 27.7 | 33.0 | 5.3 |
| $C$ | [3, 3, 3, 3, 1] | [2, 2, 2, 2, 1] | [1, 1, 1, 1, 0] | [48, 96, 192, 384, 384] | 435 | 1.0 | 1.2 | **24.0** | **24.7** | **0.7** |
| $S1$ | [3, 3, 3, 2, 1] | [2, 2, 2, 2, 1] | [1, 1, 1, 0, 0] | [42, 104, 208, 416, 384] | 422 | 0.8 | 1.3 | 24.3 | 25.1 | 0.8 |
| $S2$ | [3, 3, 3, 4, 1] | [2, 2, 1, 4, 1] | [1, 1, 1, 0, 0] | [32, 64, 128, 256, 384] | 422 | 0.7 | 1.1 | 24.3 | 25.3 | 1.0 |
| $S3$ | [3, 3, 3, 8, 1] | [2, 1, 1, 8, 1] | [1, 1, 1, 0, 0] | [17, 34, 68, 136, 384] | 458 | 0.7 | 1.6 | 25.1 | 26.2 | 1.1 |
| $S4$ | [3, 3, 3, 16, 1] | [1, 1, 1, 16, 1] | [1, 1, 1, 0, 0] | [8, 16, 32, 64, 384] | 407 | 0.6 | 2.9 | 26.2 | 27.9 | 1.3 |

Table 3: **Stem designs**: We compare ViT's standard patchify stem ($P$) and our convolutional stem ($C$) to four alternatives ($S1$ - $S4$) that each include a *patchify layer*, *i.e.*, a convolution with kernel size ($> 1$) equal to stride (highlighted in blue). Results use 50 epoch training, 4GF model size, and optimal *lr* and *wd* values for all models. We observe that increasing the pixel size of the patchify layer ($S1$ - $S4$) systematically degrades both top-1 error and optimizer stability ($\Delta$) relative to $C$.

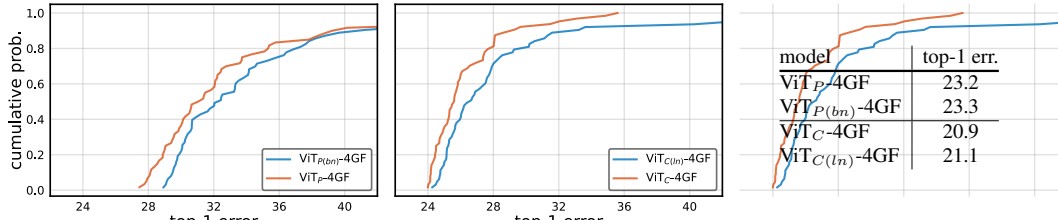

| model | top-1 err. |
|-------|------------|
| ViT$_P$-4GF | 23.2 |
| ViT$_{P(bn)}$-4GF | 23.3 |
| ViT$_C$-4GF | 20.9 |
| ViT$_{C(ln)}$-4GF | 21.1 |

Figure 7: **Stem normalization and non-linearity**: We apply BN and ReLU after the patchify stem and train ViT$_P$-4GF (*left plot*), or replace BN with layer norm (LN) in the convolutional stem of ViT$_C$-4GF (*middle plot*). EDFs are computed by sampling *lr* and *wd* values and training for 50 epochs. The table (*right*) shows 100 epoch results using best *lr* and *wd* values found at 50 epochs. The minor gap in error in the EDFs and at 100 epochs indicates that these choices are fairly insignificant.

## Appendix A: Stem Design Ablation Experiments

ViT's patchify stem differs from the proposed convolutional stem in the type of convolution used and the use of normalization and a non-linear activation function. We investigate these factors next.

**Stem design.** The focus of this paper is studying the large, positive impact of changing ViT's default patchify stem to a simple, standard convolutional stem constructed from stacked stride-two $3\times3$ convolutions. Exploring the stem design space, and more broadly "hybrid ViT" models [13], to maximize peak performance is an explicit *anti-goal* because we want to study the impact under minimal modifications. However, we can gain additional insight by considering alternative stem designs that fall between the patchify stem ($P$) the standard convolutional stem ($C$). Four alternative designs ($S1$ - $S4$) are presented in Table 3. The stems are designed so that overall model flops remain comparable. Stem $S1$ modifies $C$ to include a small $2\times2$ patchify layer, which slightly worsens results. Stems $S2$ - $S4$ systematically increase the pixel size $p$ of the patchify layer from $p = 2$ up to 16, matching the size used in stem $P$. *Increasing $p$ reliably degrades both error and optimizer stability.* Although we selected the $C$ design *a priori* based on existing best-practices for CNNs, we see *ex post facto* that it outperforms four alternative designs that each include one patchify layer.

**Stem normalization and non-linearity.** We investigate normalization and non-linearity from two directions: (1) adding BN and ReLU to the default patchify stem of ViT, and (2) changing the normalization in the proposed convolutional stem. In the first case, we simply apply BN and ReLU after the patchify stem and train ViT$_P$-4GF (termed ViT$_{P(bn)}$-4GF) for 50 and 100 epochs. For the second case, we run four experiments with ViT$_C$-4GF: {50, 100} epochs $\times$ {BN, layer norm (LN)}. As before, we tune *lr* and *wd* for each experiment using the 50-epoch schedule and reuse those values for the 100-epoch schedule. We use AdamW for all experiments. Figure 7 shows the results. From the EDFs, which use a 50 epoch schedule, we see that the addition of BN and ReLU to the patchify stem slightly worsens the best top-1 error but does not affect *lr* and *wd* stability (*left*). Replacing BN with LN in the convolutional stem marginally degrades both best top-1 error and stability (*middle*). The table (*right*) shows 100 epoch results using optimal *lr* and *wd* values chosen from the 50 epoch runs. At 100 epochs the error gap is small indicating that these factors are likely insignificant.

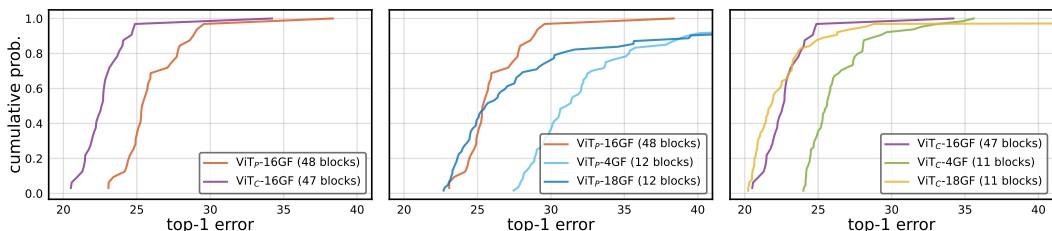

Figure 8: **Deeper models**: We increase the depth of ViT$_P$-4GF from 12 to 48 blocks, termed as ViT$_P$-16GF (48 blocks), and create a counterpart with a convolutional stem, ViT$_C$-16GF (47 blocks); all models are trained for 50 epochs. *Left*: The convolutional stem significantly improves error and stability despite accounting for only ∼2% total flops. *Middle*, *Right*: The deeper 16GF ViTs clearly outperform the shallower 4GF models and achieve similar (slightly worse) error to the shallower and wider 18GF models. The deeper ViT$_P$ also has better *lr/wd* stability than the shallower ViT$_P$ models.

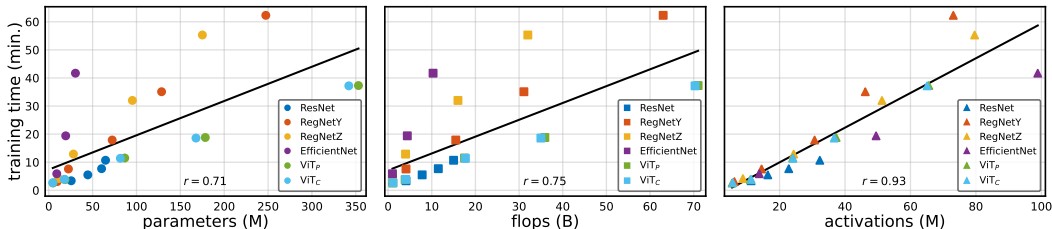

Figure 9: **Complexity measures *vs*. runtime**: We plot the GPU runtime of models versus three commonly used complexity measures: *parameters*, *flops*, and *activations*. For all models, including ViT, *runtime is most correlated with activations*, not flops, as was previously shown for CNNs [12].

## Appendix B: Deeper Model Ablation Experiments

Touvron *et al.* [42] found that deeper ViT models are more unstable, *e.g.*, increasing the number of transformer blocks from 12 to 36 may cause a ∼10 point drop in top-1 accuracy given a fixed choice of *lr* and *wd*. They demonstrate that stochastic depth and/or their proposed LayerScale can remedy this training failure. Here, we explore deeper models by looking at EDFs created by sampling *lr* and *wd*. We increase the depth of a ViT$_P$-4GF model from 12 blocks to 48 blocks, termed ViT$_P$-16GF (48 blocks). We then remove one block and use the convolutional stem from ViT$_C$-4GF, yielding a counterpart ViT$_C$-16GF (47 blocks) model. Figure 8 shows the EDFs of the two models and shallower models for comparison, following the setup in §5.3. Despite the convolutional stem accounting for only 1/48 (∼2%) total flops, it shows solid improvement over its patchify counterpart. We find that a variety of *lr* and *wd* choices allow deeper ViT models to be trained without a large drop in top-1 performance and without additional modifications. In fact, the deeper ViT$_P$-16GF (48 blocks) has better *lr* and *wd* stability than ViT$_P$-4GF and ViT$_P$-18GF over the sampling range (Figure 8, *middle*).

## Appendix C: Larger Model ImageNet-21k Experiments

In Table 2 we reported the peak performance of ViT models on ImageNet-21k up to 36GF. To study larger models, we construct a 72GF ViT$_P$ by using 22 blocks, 1152 hidden size, 18 heads, and 4 MLP multiplier. For ViT$_C$-72GF, we use the same $C$-stem design used for ViT$_C$-18GF and ViT$_C$-36GF, but *without* removing one transformer block since the flops increase from the $C$-stem is marginal in this complexity regime.

Our preliminary explorations into 72GF ViT models directly adopted hyperparameters used for 36GF ViT models. Under this setting, we observed that the convolutional stem still improves top-1 error, however, we also found that a new form of instability arises, which causes training error to randomly spike. Sometimes training may recover within the same epoch, and subsequently the final accuracy is not impacted; or, it may take several epochs to recover from the error spike, and in this case we observe suboptimal final accuracy. The first type of error spike is more common for ViT$_P$-72GF, while the latter type of error spike is more common for ViT$_C$-72GF.

| model | AdamW | | SGD | |
|---|---|---|---|---|
| | *lr* | *wd* | *lr* | *wd* |
| RegNetY-* | 3.8e-3 | 0.1 | 2.54 | 2.4e-5 |
| $\text{ViT}_P$-1GF | 2.0e-3 | 0.20 | 1.9 | 1.3e-5 |
| $\text{ViT}_P$-4GF | 2.0e-3 | 0.20 | 1.9 | 1.3e-5 |
| $\text{ViT}_P$-18GF | 1.0e-3 | 0.24 | 1.1 | 1.2e-5 |
| $\text{ViT}_C$-1GF | 2.5e-3 | 0.19 | 1.9 | 1.3e-5 |
| $\text{ViT}_C$-4GF | 1.0e-3 | 0.24 | 1.3 | 2.2e-5 |
| $\text{ViT}_C$-18GF | 1.0e-3 | 0.24 | 1.1 | 2.7e-5 |

| model | AdamW | |
|---|---|---|
| | *lr* | *wd* |
| ViT-* | $(2.5e{-}4, 8.0e{-}3)$ | $(0.02, 0.8)$ |
| RegNetY-* | $(1.25e{-}3, 4.0e{-}2)$ | $(0.0075, 0.24)$ |

| model | SGD | |
|---|---|---|
| | *lr* | *wd* |
| ViT-* | $(0.1, 3.2)$ | $(4.0e{-}6, 1.2e{-}4)$ |
| RegNetY-* | $(0.25, 8.0)$ | $(3.0e{-}6, 8.0e{-}5)$ |

Table 4: **Learning rate and weight decay used in §5**: *Left*: Per-model *lr* and *wd* values used for the experiments in §5.1 and §5.2, optimized for ImageNet-1k at 50 epochs. *Right*: Per-model *lr* and *wd* ranges used for the experiments in §5.3. Note that for our final experiments in §6, we constrained the *lr* and *wd* values further, using a single setting for all CNN models, and just two settings for all ViT models. We recommend using this simplified set of values in §6 when comparing models for fair and easily reproducible comparisons. All *lr* values are normalized w.r.t. a minibatch size of 2048 [16].

To mitigate this instability, we adopt two measures: (i) For both models, we lower *wd* from $0.28$ to $0.15$ as we found that it significantly reduces the chance of error spikes. (ii) For $\text{ViT}_C$-72GF, we initialize its stem from the ImageNet-21k pre-trained $\text{ViT}_C$-36GF and keep it frozen throughout training. These modifications make training ViT-72GF models on ImageNet-21k feasible. When fine-tuned on ImageNet-1k, $\text{ViT}_P$-72GF reaches $14.2\%$ top-1 error and $\text{ViT}_C$-72GF reaches $13.6\%$ top-1 error, showing that $\text{ViT}_C$ still outperforms its $\text{ViT}_P$ counterpart. Increasing fine-tuning resolution from $224$ to $384$ boosts the performance of $\text{ViT}_C$-72GF to $12.6\%$ top-1 error, while significantly increasing the fine-tuning model complexity from 72GF to 224GF.

## Appendix D: Model Complexity and Runtime

In previous sections, we reported error *vs.* training time. Other commonly used complexity measures include *parameters*, *flops*, and *activations*. Indeed, it is most typical to report accuracy as a function of model flops or parameters. However, flops may fail to reflect the bottleneck on modern memory-bandwidth limited accelerators (*e.g.*, GPUs, TPUs). Likewise, parameters are an even more unreliable predictor of model runtime. Instead, activations have recently been shown to be a better proxy of runtime on GPUs (see [12, 31]). We next explore if similar results hold for ViT models.

For CNNs, previous studies [12, 31] defined *activations* as the *total size of all output tensors of the convolutional layers*, while disregarding normalization and non-linear layers (which are typically paired with convolutions and would only change the activation count by a constant factor). In this spirit, for transformers, we define *activations as the size of output tensors of all matrix multiplications*, and likewise disregard element-wise layers and normalizations. For models that use both types of operations, we simply measure the output size of all convolutional and vision transformer layers.

Figure 9 shows the runtime as a function of these model complexity measures. The Pearson correlation coefficient ($r$) confirms that activations have a much stronger linear correlation with actual runtime ($r = 0.93$) than flops ($r = 0.75$) or parameters ($r = 0.71$), confirming that the findings of [12] for CNNs also apply to ViTs. While flops are somewhat predictive of runtime, models with a large ratio of activations to flops, such as EfficientNet, have much higher runtime than expected based on flops. Finally, we note that $\text{ViT}_P$ and $\text{ViT}_C$ are nearly identical on all complexity measures and runtime.

**Timing.** Throughout the paper we report *normalized* training time, as if the model were trained on a single 8 V100 GPU server, by multiplying the actual training time by the number of GPUs used and dividing by 8. (Due to different memory requirements of different models, we may be required to scale up the number of GPUs to accommodate the target minibatch size.) We use the number of minutes taken to process one ImageNet-1k epoch as a standard unit of measure. We prefer training time over inference time because inference time depends heavily on the use case (*e.g.*, a streaming, latency-oriented setting requires a batch size of 1 *vs.* a throughput-oriented setting that allows for batch size $\gg 1$) and the hardware platform (*e.g.*, smartphone, accelerator, server CPU).

| model | Augment | Mixup | CutMix | Label Smooth | Model EMA | Erasing | Stoch Depth | Repeating | 100 epochs | 400 epochs | 300 epochs [41] |
|---|---|---|---|---|---|---|---|---|---|---|---|
| | Auto | ✓ | ✓ | ✓ | ✓ | | | | 23.2 | 20.5 | - |
| | Rand | ✓ | ✓ | ✓ | | | ✓ | ✓ | ✓ | 25.4 | 20.7 | - |
| ViT$_P$-4GF | Rand | ✓ | ✓ | ✓ | | | ✓ | | ✓ | 24.9 | 20.5 | - |
| | Rand | ✓ | ✓ | ✓ | | | ✓ | | | 23.6 | 20.4 | - |
| | Rand | ✓ | ✓ | ✓ | | | | | | 23.5 | 20.3 | - |
| | Auto | ✓ | ✓ | ✓ | | | | | | 23.0 | 20.3 | - |
| | Auto | ✓ | ✓ | ✓ | ✓ | | | | 19.9 | 17.9 | - |
| | Rand | ✓ | ✓ | ✓ | | | ✓ | ✓ | ✓ | 22.5 | 18.6 | 18.2 |
| | Rand | ✓ | ✓ | ✓ | | | ✓ | | ✓ | 25.1 | 19.2 | 96.6 |
| | Rand | ✓ | ✓ | ✓ | | | ✓ | | | 21.2 | 19.9 | - |
| ViT$_P$-18GF | Rand | ✓ | ✓ | ✓ | | | | | | 20.9 | 19.7 | - |
| | Auto | ✓ | ✓ | ✓ | | | | | | 20.4 | 20.0 | - |
| | Rand | ✓ | ✓ | ✓ | | | ✓ | ✓ | | - | - | 22.6 |
| | Rand | ✓ | ✓ | ✓ | | | | ✓ | ✓ | - | - | 95.7 |
| | Rand | ✓ | ✓ | ✓ | ✓ | ✓ | ✓ | ✓ | ✓ | - | - | 18.1 |

Table 5: **Ablation of data augmentation and regularization**: We use the *lr* and *wd* from Table 4 (left), except for ViT$_P$-18GF models with RandAugment which benefit from stronger *wd* (we increase *wd* to 0.5). Original DeiT ablation results [41] are copied for reference in gray (*last column*); these use a *lr/wd* of $1e-3/0.05$ (*lr* normalized to minibatch size 2048), which leads to some training failures (we note our *wd* is 5-10× higher). Our default training setup (*first row* in each set) uses AutoAugment, mixup, CutMix, label smoothing, and model EMA. Compared to the DeiT setup (*second row* in each set), we do not use erasing, stochastic depth, or repeating. Although our setup is equally effective, it is simpler and also converges much faster (see Figure 10).

## Appendix E: Additional Experimental Details

**Stability experiments.** For the experiments in §5.1 and §5.2, we allow each CNN and ViT model to select a different *lr* and *wd*. We find that all CNNs select nearly identical values, so we normalize them to a single choice as done in [12]. ViT models prefer somewhat more varied choices. Table 4 (*left*) lists the selected values. For the experiments in §5.3, we use *lr* and *wd* intervals shown in Table 4 (*right*). These ranges are constructed by (i) obtaining initial good *lr* and *wd* choices for each model family; and then (ii) multiplying them by $1/8$ and 4.0 for left and right interval endpoints (we use an asymmetric interval because models are trainable with smaller but not larger values). Finally we note that if we were to redo the experiments, the setting used in §5.1/§5.2 could be simplified.

**Peak performance on ImageNet-1k.** We note that in later experiments we found tuning *lr* and *wd* per model is *not* necessary to obtain competitive results. Therefore, for our final experiments in §6, we constrained the *lr* and *wd* values further, using a single setting for all CNN models, and just two settings for all ViT models, as discussed in §6. We recommend using this simplified set of values when comparing models for fair and easily reproducible comparisons. Finally, for these experiments, when training is memory constrained (*i.e.*, for EfficientNet-{B4,B5}, RegNetZ-{4,16,32}GF), we reduce the minibatch size from 2048 and linearly scale the *lr* according to [16].

**Peak performance on ImageNet-21k.** For ImageNet-21k, a dataset of 14M images and ~21k classes, we pretrain models for 90 (ImageNet-21k) epochs, following [13]. We do *not* search for the optimal settings for ImageNet-21k and instead use the identical training recipe (up to minibatch size) used for ImageNet-1k. To reduce training time, we distribute training over more GPUs and use a larger minibatch size of 4096 with the *lr* scaled accordingly. For simplicity and reproducibility, we use a single label per image, unlike some prior work (*e.g.*, [35, 40]) that uses WordNet [28] to expand single labels to multiple labels. After pretraining, we fine-tune for 20 epochs on ImageNet-1k and use a small-scale grid search of *lr* while keeping *wd* at 0, similar to [13, 40].

## Appendix F: Regularization and Data Augmentation

At this study's outset, we developed a simplified training setup for ViT models. Our goals were to design a training setup that is as simple as possible, resembles the setup used for state-of-the-art CNNs [12], and maintains competitive accuracy with DeiT [41]. Here, we document this exploration

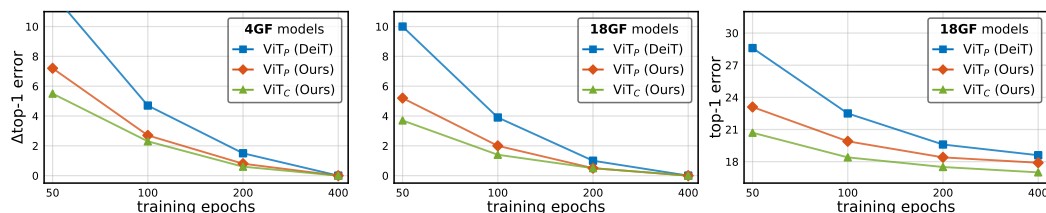

Figure 10: **Impact of training recipes on convergence**: We train ViT models using the DeiT recipe *vs.* our simplified counterpart. *Left and middle*: $\Delta$top-1 error of 4GF and 18GF models at 50, 100 and 200 epoch schedules, and asymptotic performance at 400 epochs. *Right*: Absolute top-1 error of 18GF models. Removing augmentations and using model EMA accelerates convergence for both $\text{ViT}_P$ and $\text{ViT}_C$ models while slightly improving upon our reproduction of DeiT's top-1 error.

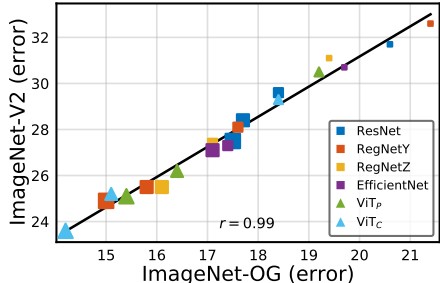

Figure 11: **ImageNet-V2 performance**: We take the models from Table 2 and benchmark them on the ImageNet-V2 test set. Top-1 errors are plotted for the original (OG) ImageNet validation set (x-axis) and the ImageNet-V2 test set (y-axis). Rankings are mostly preserved up to one standard deviation of noise (estimated at ∼0.1-0.2%) and the two testing sets exhibit linear correlation (Pearson's $r = 0.99$). Marker size corresponds to model flops.

by considering the baseline $\text{ViT}_P$-4GF and $\text{ViT}_P$-18GF models. Beyond simplification, we also observe that our training setup yields faster convergence than the DeiT setup, as discussed below.

Table 5 compares our setup to that of DeiT [41]. Under their *lr/wd* choice, [41] report failed training when removing *erasing* and *stochastic depth*, as well as significant drop of accuracy when removing *repeating*. We find that they can be safely disabled as long as a higher *wd* is used (our *wd* is 5-10× higher). We observe that we can remove model EMA for $\text{ViT}_P$-4GF, but that it is essential for the larger $\text{ViT}_P$-18GF model, especially at 400 epochs. Without model EMA, $\text{ViT}_P$-18GF can still be trained effectively, but this requires additional augmentation and regularization (as in DeiT).

Figure 10 shows that our training setup accelerates convergence for both $\text{ViT}_P$ and $\text{ViT}_C$ models, as can be seen by comparing the error *deltas* ($\Delta$top-1) between the DeiT baseline and ours (*left* and *middle* plots). Our training setup also yields slightly better top-1 error than our reproduction of DeiT (*right* plot). We conjecture that faster convergence is due to removing repeating augmentation [1, 20], which was shown in [1] to slow convergence. Under some conditions repeating augmentation may improve accuracy, however we did not observe such improvements in our experiments.

## Appendix G: ImageNet-V2 Evaluation

In the main paper and previous appendix sections we benchmarked all models on the original (OG) ImageNet validation set [10]. Here we benchmark our models on the ImageNet-V2 [33], a new test set collected following the original procedure. We take the 400-epoch *or* ImageNet-21k models from Table 2, depending on which one is better, and evaluate them on ImageNet-V2 to collect top-1 errors. Figure 11 shows that rankings are mostly preserved up to one standard deviation of noise (estimated at ∼0.1-0.2%). The two testing sets exhibit linear correlation, as confirmed by the Pearson correlation coefficient $r = 0.99$, despite ImageNet-V2 results showing higher absolute error. The parameters of the fit line are given by $y = 1.31x + 5.0$.