# OpenReview forum: "Early Convolutions Help Transformers See Better"
_NeurIPS.cc/2021/Conference — NeurIPS 2021 Poster_

### Official Review · Reviewer_vj7C · 2021-07-15

**Rating:** 7
**Confidence:** 4

**Summary:**

The paper studies the impact of stem in ViT. Specifically, the authors propose to replace the patch-based stem in ViT with a standard, lightweight convolutional stem, and show that this simple revision leads to substantially improved stability wrt the choices of optimizers and several other hyper-parameters. The resulting model achieves comparable performance with strong convnets baselines when trained on ImageNet-1K and has better accuracy-speed tradeoff than both ViTs and convents when trained over larger dataset (ImageNet-21K).

**Limitations And Societal Impact:**

The authors mentioned "understanding the theoretical foundation of why such a minimal architectural modification can have such large (positive) impact on optimizability" as a future work. This is actually a major missing piece IMO, and I believe answering this question in future revisions could substantially strengthen the technical contribution of this work.

**Main Review:**

Strengths:
+ Overall, the paper is very well written and easy to follow.
+ Extensive amount of ablation studies have been provided to back up the main claim that the conv stem can improve ViT's optimization stability/robustness.
+ IMO the most interesting part of the manuscript is not the resulting model itself, but rather the methodology. For example, the notion of "optimizability" examined in the paper can be potentially used as an additional metric to evaluate new architectures and provide complementary insights other than accuracy or speed.

Weaknesses:
- The overall message is interesting but not strong enough to be exciting:
  * In most ablation studies, ViT_c behaves like something in the middle of ViT and ConvNets (e.g., see Figure 2, Figure 3 and Figure 5). Those results are interesting, but on the other hand are more or less expected given the hybrid nature of the model.
  * More importantly, although the authors provided extensive empirical evidence to back up their claim that a conv stem is helpful, the critical question that why it is effective is left unanswered. An in-depth analysis of the underlying reason for those empirical observations would make the paper a lot more informative.
- The paper claims their model outperforms the state-of-the-art convnets, but I'm not sure whether it is true: while RegNet is a competitive baseline, there are more recent networks with stronger performance (e.g., NFNets).
- There are missing details about the ImageNet-21K training setup that I cannot find in paper or appendix. E.g., what is the input resolution during fine-tuning (224 or 384)?

Overall, I'm leaning positive about the paper because the methodology is interesting and I believe the data points would be useful for future studies. On the negative side, not much insight was provided to answer why conv stems are helpful, and I believe answering that technical question would substantially strengthen the work.

========= post rebuttal ==========

Raising my rating to 7 since some of my technical questions have been addressed in the author's response.

**Time Spent Reviewing:**

3

---

> ### Author Response · Authors · 2021-08-09
> **The Authors' Response to Reviewer vj7C**
>
> We thank the reviewer for the constructive feedback. We are delighted that the reviewer affirms the methodology used in our work and would like to respond to the concerns.
>
> 1. *Why convolution stem is helpful is left unanswered in the paper.*
> - After we discovered that the convolution stem was helpful, we conducted several studies in an attempt to explain why this is the case. For example, we’ve gathered supporting evidence showing that shift-invariance (at initialization) is correlated with accuracy. **However, after extensive discussion when we were drafting the paper, we came to the conclusion that this result could only show a correlation and was not strong enough to provide a satisfying answer. We feel that our work in itself is valuable, and that in the deep learning era, it is prudent to write a scientific explanation only when we have sufficient evidence.**
>
> - We would like to take the original Batch Norm paper (Ioffe and Szegedy, 2015) as an example: it has a technique that is incredibly effective and has been one of the most fundamental publications for training deep neural networks, but its explanation (reducing internal covariate shift) is actually wrong [Santurkar et al, NeurIPS 2018]. It would later take many papers to gradually discover why BN is so effective. In our case, the convolutional stem is effective as well, and better theories to uncover the underlying reason would come later.
>
> 2. *ViT_C behaves like something in the middle of ViT(P) and ConvNets*
> - The review is completely correct in the figures that the reviewer refers to, i.e., Fig. 2, 3 and 5. However, it is only one part of the larger picture - these figures are for measuring the **stability** of ViT models, hence they only reflect the deltas (gaps) of model performance. As reviewer VuQn pointed out, fig 6 reveals the tremendous benefits of ViT_C when it is scaled to a larger dataset such as ImageNet-21k. Therefore ViT_C is not in the middle of ViT(P) and ConvNets.
>
> - We’d also like to note, as discussed in the introduction section of our paper, that using only a few convolutional layers in the early visual processing makes a big difference. As the convolutional layers account for ~1/12 = 8% of the model’s complexity, it is *rather unexpected*.
>
> 3. *RegNet is a competitive baseline, there are more recent networks with stronger performance (e.g., NFNets).*
> - We’d like to note that NFNets was uploaded onto the non-peer reviewed arXiv preprint server in mid February, while the NeurIPS submission deadline is in May. Moreover, NFNet requires meticulously designed training strategies, such as gradient clipping (see Section 5), to compensate for the absence of batch normalizations. RegNet, on the other hand, uses a **standard training recipe** (important for our stability analysis) and **achieves very good accuracy**. Ergo we choose RegNet as an anchor to study the optimizability of state-of-the-art ConvNets.
>
> 4. *ImageNet-21k finetuning resolutions (224 or 384).*
> - For all ViT experiments we use 224x224 and our results could be further improved using techniques like higher resolution or distillation, but we felt this was a distraction from the focus of the paper, and would lead to higher flops.

---

> > ### Comment · Reviewer_vj7C · 2021-08-27
> > **Thank you for the response**
> >
> > Since some of my technical questions have been addressed in the author's response, I’d like to raise my rating to 7. Some further comments below:
> >
> > I agree that not providing an answer is better than providing a misleading one. For the exact same reason, the batch norm paper is probably a bad example. While it makes sense not to cite arXiv preprints in this submission, I also believe "not using standard training techniques" can be a highly subjective judgement and does not feel like a scientific enough excuse to ignore an existing work. To me a more convincing option is to cover the literature comprehensively as possible and then clearly explain in the paper why RegNet is a good enough baseline for this study (as detailed in the author’s response). IMO doing this would only strengthen this work further.

---

### Official Review · Reviewer_VuQn · 2021-07-16

**Rating:** 9
**Confidence:** 5

**Summary:**

The authors perform an extensive study investigating the stem of ViT models and replacing it with one that consists of a stack of downsampling 3x3 convolutions. The study reveals that this simple change makes ViT training more robust, faster to converge, but also scales better (cf. i21k pre-training).

**Main Review:**

I have to admit, that I was initially set to reject the paper, because I have counter-evidence of the main claim that "ViT cannot be optimized by sgdm": I have been training original patch-stem ViT models with SGDM for a while, to same accuracy as AdamW (but unpublished), and am certain that the negative results reported in DeiT are due to confounders, such as not properly tuning sgdm.

However, carefully reading the paper made me change my mind by 180°. This paper is really good and needs to be accepted. Many statements reflect my thoughts:
- It is crucial to tune lr/wd, or one gets misleading results such as "random erasing is necessary", and actually tuning those reveals it (and other tricks) to be *unnecessary*.
- Especially for ViT and EfficientNet models, flops and #params are not a good measurement to put on the X axis.
- We need to compare architectures in ideal settings for each.

I find the very last presented finding, Fig6(right), to be the most interesting: the ViT_C seems to scale better than both ViT_P and Re{s/g}Nets. This is very promising.

Now some criticism on the paper:

1) line 112: I would argue that the conv-stem is not "strictly beneficial" since:
1) A) the patchify stem has led to some interesting recent ideas which are simply not possible with a conv-stem, such as using different random augmentations per patch.
1) B) It re-introduces BatchNorm into the model, with all of its problems. I wish authors had tried using GroupNorm here too.
2) line 94: the cited papers do not really support "optimization difficulty".
3) Why do we need the final 1x1 conv in the stem? Couldn't we simply have the last 3x3 conv output the required number of channels? That would furhter simplify the design.
4) line 139: "we do not optimize the stem to maximize model accuracy". This is not really true to spirit, given AppendixA. Even if the best stem ends up being the one that authors first thought of. What if a different one from AppendixA was found to perform substantially better? Surely, that one would have been proposed instead. Simply drop this sentence.
5) Reusing optimal lr/wd from 50ep for up to 400ep. This is known to be suboptimal (cf AdamW paper itself) and generally for much longer training, wd needs to be decreased. It likely does not invalidate the results though.
6) It is not explicitly stated what lr/wd are used in 5.1 and 5.2. Is it the best ones found in 5.3? If not, this might be an issue.
7) Does the SGDM sweep also use "decoupled" WD as per AdamW? Do the authors actually use decoupled WD from AdamW? (That is not commonly used in practice, but better for such sweeps.) By "decoupled", I mean doing w_t+1 = w_t - lr * grad w_t - wd * w_t,  as opposed to w_t+1 = w_t - lr * grad w_t - wd * lr * w_t, see also the AdamW paper.

**Time Spent Reviewing:**

3

---

> ### Author Response · Authors · 2021-08-09
> **The Authors' Response to Reviewer VuQn**
>
>
> We thank the reviewer for the constructive feedback. We share the reviewer’s frustration for some of the bizarre observations in previous works, very likely due to confounders, and appreciate the reviewer’s strong support for the merit of our work. We believe, as the reviewer, that our work would contribute to not only the debate of conv-free architectures, but also the scientific methodology for investigating and benchmarking neural network architectures.
>
> We agree with the reviewer on many of the improvements that can be made for our draft. A few clarifications:
>
> 1. *Why do we need the final 1x1 conv in the stem? Couldn't we simply have the last 3x3 conv output the required number of channels? That would further simplify the design.*
> - We fully agree with the spirit of this suggestion, but use 1x1 conv for pragmatic reasons. We would like the conv stem to account for ~1 ViT block flops. If we use 3x3 conv to match the ViT input channels, the last two conv layers would be (3x3 conv, channel k1) -> (3x3 conv, channel ViT backbone), which is of much higher flops compared to (3x3 conv, channel k1) -> (1x1 conv, channel ViT backbone). The 1x1 conv design avoids allocating too many flops to the last 3x3 conv in the stem.
>
> 2. *It is not explicitly stated what lr/wd are used in 5.1 and 5.2. Is it the best ones found in 5.3? If not, this might be an issue.*
> - Yes the lr/wd in section 5.1 and 5.2 are the best ones found in section 5.3. We will make it more clear in the next version.
>
> 3. *Clarification of SGDM and AdamW*
> - We used the default SGDM and AdamW implementations in PyTorch, thus for AdamW, the update uses - wd * lr * w_t. We will clarify this detail in the paper. Thank you for the suggestion to explore the alternative update rule.
>
> 4. *Batch norm vs layer norm in the conv stem*
> - We tried LN to replace BN in the conv stem (see Appendix). The preliminary results look promising and we like the suggestion to also investigate Group Norm.
>
> 5. *“Strictly beneficial”*
> - We agree with the reviewer that it sounds too strong, and we will revise it accordingly.

---

> > ### Comment · Reviewer_VuQn · 2021-08-10
> > **Thanks**
> >
> > Thank you for the answer. From this perspective, the final 1x1 makes a lot of sense.
> >
> > Having read all the other reviews and author answers, I will stick to my strong acceptance recommendation. Keep up the good work.

---

### Official Review · Reviewer_G3R8 · 2021-07-18

**Rating:** 7
**Confidence:** 4

**Summary:**

This paper overall tackles a very important question and pin-points the important dissociation that when the computational complexity of two models is equalized: one with a first stage convolutional operator and another one without one -- both pre-seeded to Vision Transformers; it is the one with a convolutional stem/operator *that is only sufficiently necessary in the first layer!* that exploits the locality structure in image, has increase in accuracy, faster convergence and flexible optimizability.


**Main Review:**


Strengths:

This paper presents all the correct ingredients for acceptance at any computer vision venue -- despite this being NeurIPS: The paper presents a strong set of experiments, baselines, and several simulations ran for different conditions (example GFlops, and hidden units in the ViT’s) that have been carefully tuned to show the stability of the results despite a lack of errorbars. Overall this paper should be accepted because it presents a simple idea that has been rigorously tested through various experimental manipulations and the novelty is presented quite clearly in Figure 1. The take-home (See above) is easy: if you want *better* results (accuracy/faster convergence/flexible optimizability) when training a Vision Transformer, add a convolutional stem/layer in the early block of visual processing (somewhat surprisingly similar -- to how the human visual system operates in early stages of processing).

I have no follow-up questions to the authors, but I would like to add some limitations that prevent me from increasing my score (that is already very positive), as this paper could have had potential to have a more interdisciplinary scope beyond the Transformer movement.

Limitations:

There is more to computer vision (and vision in general) than object recognition and ImageNet-based experiments. Several follow-up questions arise, such as: How would a convolutional-stem aid a perceptual system if the task was *not* object recognition (the title of ‘seeing’, suggests Object Recognition is the ultimate standard). But more critically, experiments with Robustness to either common corruptions (Hendrycks & Dietterich, ICLR 2019) or Adversarial attacks like any of the PGD-flavors should be explored in future work. Not to be nosey, but these are indeed important questions as several authors have shown that there is a trade-off between accuracy and robustness (Zhang et al., ICML 2019).

I do wish that the paper adds more discussion on the ML theoretical aspects of the advantages of convolution when a locality constraint is induced by the image structure as explored in Poggio (PNAS 2020) and recently, Deza, Banburski, Liao & Poggio (ArXiv 2021). Similarly, it would be nice if the paper connects the importance of the convolutional operator as a biologically plausible inductive bias (eg. V1 gabor-like filters; Hubel & Wiesel in the 60’s) and how recent trends in computer vision have started to revisit the importance of inductive biases in 2-stage *neuroscience-inspired* perceptual systems as the work of Dapello, Marques et al. for the case of Adversarial Robustness (NeurIPS, 2020); Parthasarathy & Simoncelli (ArXiv, 2020) for texture recognition w/ a self-supervised objective and Deza & Konkle (ArXiv, 2020) for scene recognition w/ a foveated module. Perhaps adding a small section in the Relevant work about these and other works that have explored the importance of locality, convolutional operators in CNNs and 2-stage models should be addressed.

Other works worth adding that are particularly relevant to the theory of locality and convolutional operators as a critical computation in machines:

* Elsayed et al. “Revisiting Spatial Invariance with Low-Rank Local Connectivity”. ICML 2020.

* Pogodin et al.: “Towards Biologically Plausible Convolutional Networks”. ArXiv 2021.

* d’Ascoli et al: “Finding the Needle in the Haystack with Convolutions: on the benefits of architectural bias” NeurIPS 2019.


**Time Spent Reviewing:**

3

---

> ### Author Response · Authors · 2021-08-09
> **The Authors' Response to Reviewer G3R8**
>
> We thank the reviewer for the constructive feedback. The references provided by the reviewer are thorough, informative and inspiring. Accordingly, we will add the discussion into the related work section in the next version of the paper.
>
> We agree with the reviewer that robustness is an important factor along with a model's accuracy. It is a good suggestion and we believe that the results would be interesting to see.

---

### Decision · Program_Chairs · 2021-09-27

**Decision:**

Accept (Poster)

**Comment:**

Reviewers agree after author's response that this paper is a (strong) accept (7,9,7). The paper doesn't have any major weak points. The message is simple and clear and the contribution thorough. I don't consider the implications of the result worthy of a spotlight though.